# Field-Crop Soils in Eastern France: Coldspots of Azole-Resistant *Aspergillus fumigatus*

**DOI:** 10.3390/jof9060618

**Published:** 2023-05-27

**Authors:** Chloé Godeau, Nadia Morin-Crini, Grégorio Crini, Jean-Philippe Guillemin, Anne-Sophie Voisin, Sylvie Dousset, Steffi Rocchi

**Affiliations:** 1Chrono-Environnement UMR6249, CNRS Franche-Comté University, 25000 Besançon, France; 2Agroécologie, INRAE, Institut Agro, Bourgogne University, 21000 Dijon, France; 3Laboratoire Interdisciplinaire des Environnements Continentaux, UMR 7360 Lorraine University/CNRS, 54506 Vandoeuvre lès Nancy, France; 4Parasitology-Mycology Department, University Hospital of Besançon, 25000 Besançon, France; 5Smaltis, Bioinnovation, 4 Rue Charles Bried, 25000 Besançon, France

**Keywords:** *Aspergillus fumigatus*, triazole, coldspot, resistance, field crop, qPCR

## Abstract

Triazole fungicides are widely used to treat fungal pathogens in field crops, but very few studies have investigated whether fields of these crops constitute hotspots of azole resistance in *Aspergillus fumigatus*. Soil samples were collected from 22 fields in two regions of eastern France and screened for triazole residues and azole-resistant *A. fumigatus* (ARAf). Real-time quantitative PCR (qPCR) was used to quantify *A. fumigatus* in these soil samples. All the plots contained tebuconazole at concentrations from 5.5 to 19.1 ng/g of soil, and 5 of the 22 plots also contained epoxiconazole. Only a few fungal isolates were obtained, and no ARAf was detected. *A. fumigatus* qPCR showed that this fungal species was, on average, 5000 times more common in soil from flowerbeds containing ARAf than in soil from field crops. Thus, field-crop soils do not appear to promote *A. fumigatus* development, even if treated with azole fungicides, and cannot be considered hotspots of resistance. Indeed, our results suggest that they are instead a coldspot of resistance and highlight how little is known about the ecological niche of this species.

## 1. Introduction

*Aspergillus fumigatus* is a ubiquitous saprophytic mould. Its natural ecological niche is the soil, in which it survives and grows on organic debris. It is involved in the degradation of organic matter and is found in diverse ecological conditions [1]. It produces conidia that can be carried over long distances by wind and may constitute a health risk, as this fungal species is an opportunistic pathogen in both animals and humans [2,3]. Indeed, *A. fumigatus* is the pathogen most frequently implicated in aspergillosis, including invasive aspergillosis, which has a mortality rate of more than 50% [4].

In clinical practice, the first-line treatment for aspergillosis is azole compounds, which were released onto the market in the 1990s and have facilitated patient management. However, resistance to these molecules has been emerging over the last 20 years [5], leading some countries to recommend the avoidance of triazoles as a first-line treatment in certain patients [6]. Resistance can develop as a result of the long-term treatment of a patient, but the use of azoles in agriculture and in wood treatment has also been highlighted as an environmental source of resistance [7,8,9]. Five triazole molecules—propiconazole, bromuconazole, tebuconazole, epoxiconazole and difenoconazole—have been implicated as particularly heavily involved in the cross-resistance of *A. fumigatus* to triazoles used in human medicine [8].

Resistance mechanisms linked to mutations of the *cyp51A* gene (single-nucleotide polymorphisms) and its promoter (tandem repeats) are most commonly described in isolates acquiring resistance in the environment, but other mechanisms such as overexpression of efflux pump-encoding genes (*AtrI*; *AtrF*; *Cdr1b*) have already been observed in environmental strains [10,11,12].

Azole-resistant *A. fumigatus* (ARAf) is a growing problem, and the WHO listed this opportunistic pathogen as “critical” in October 2022. It is now important to identify the areas in which there is a risk of azole resistance developing.

An environment must satisfy two main criteria to be considered a hotspot of resistance: azole molecules must be present, together with conditions favouring the abundant and prolonged growth of *A. fumigatus*, but also reproduction and genetic variation of *A. fumigatus* [13]. Conversely, sites that do not meet these two criteria can be considered coldspots, especially those where there are low numbers of pan-azole resistant isolates [14,15]. Various products treated with azole fungicides, such as green waste, wood chips and composts, have been described as potential hotspots of resistance, capable of promoting the emergence of ARAf [13,16,17]. By contrast, few studies have focused on fields of crops treated with azole fungicides to protect crop production and quality [18]. Agricultural practices differ between countries, and there are few data concerning the frequency of ARAf as a function of the presence of triazole molecules in field crops in Europe.

The objective of this work was to determine whether large amounts of azole residues and ARAf can be detected in field crops treated with azoles and to determine whether this environment constitutes a potential hotspot of resistance.

## 2. Materials and Methods

### 2.1. Environmental Sampling

Two soil sampling campaigns were carried out in field crops in two regions of eastern France: Lorraine on 29 and 30 September, 2020 and Burgundy (near Dijon (47°13′ N, 5°03′ E)) on 27 June 2021. Crops were treated in the Lorraine region in May and June 2020 (i.e., 4 to 5 months before sampling) and in Burgundy in April and May 2021 (i.e., 1 to 2 months before sampling). A total of 22 fields (coded names BP, CBP, etc. in Table 1) were included in the study. Ten surface samples (2–10 cm depth) were collected along a transect in each field. Some of the plots were treated with azole fungicides, others had not been treated with pesticides during the year of sampling and some were farmed organically. All the fields in the Lorraine region were located within a circular area 20 km in diameter, and those in Burgundy were located within an area 5 km in diameter. The record of fungicide treatments of each field was obtained from the farmers. The characteristics of each field are presented in Table 1.

At each sampling point, two identical samples were collected: one for the analysis of azole fungicide residues and soil analyses such as granulometry, organic matter content and pH and the other for the isolation of *A. fumigatus* strains (Table 1). For physicochemical analysis, the surface samples from each plot were pooled to generate 22 composite samples for analysis. The French Aisne soil texture triangle was constructed with a soil texture package [19].

### 2.2. Physicochemical Analysis of Soils

Organic matter (OM) content was estimated by calcination. Masses were compared before and after the heating of 10 g of dried soil per plot at 550 °C for 4 h. The pH was determined with a WTR^®^ pH meter: 10 mL dry soil was shaken with 50 mL reverse-osmosis water (1:5 suspension, *v*/*v*) for 20 min. The suspension was then left to rest overnight before pH determination. Granulometric analyses were performed with an LS 230 laser diffraction particle size analyser (Beckman Coulter, Villepinte, France) to determine the proportions of clay, silt and sand for each plot and to determine soil texture.

### 2.3. Analysis of Soil Fungicide Levels

#### 2.3.1. Extraction and Analysis

Difenoconazole, propiconazole, tebuconazole, epoxiconazole and bromuconazole, which were potentially implicated in the development of environmental azole resistance in *A. fumigatus*, and a deuterated internal standard, tebuconazole d6, of analytical standard grade, were purchased from Sigma Aldrich (Saint Quentin Fallavier, France) and used without purification. Acetonitrile Distol, a range of products for organic trace analysis, was obtained from Fisher Scientific (Illkirch, France). Sodium chloride (99.5%) was purchased from Sigma Aldrich (Saint Quentin Fallavier, France).

The analytical methodology for the extraction of the five triazole fungicides in soil is based on the QuEChERS method. This technique, initially used for extracting pesticides from food, has also been developed for other environmental matrices, such as water and soils, because it is quick, easy, cheap, efficient, robust and safe [20]. Briefly, 20 mL acetonitrile was added to a suspension of 10 g of freeze-dried soil and passed through a 2 mm sieve in 5 mL of water. The mixture was vigorously stirred and 3 g sodium chloride was added to induce the separation of the aqueous phase from the organic phase. The mixture was stirred further, and the organic acetonitrile phase was then retrieved and concentrated. The five triazole molecules were then detected by gas chromatography (GC) with a 7890A GC system (Agilent Technologies, Massy, France) coupled with detection by triple quadrupole mass spectrometry (7000 GC-MS/MS model; Agilent Technologies). A 1 μL quantity of the extract was injected at a temperature of 300°C in splitless mode under a constant flow of He (purity 99.9999%) at 1.5 mL min^−1^, followed by a purge flow to split vent after 0.5 min. We used a (5% phenyl)-methylpolysiloxane HP5MS column (30 m × 0.25 mm I.D., 0.25 μm film thickness, Agilent 19091J-433). Total run time was 42.27 min. The temperatures of the transfer line, ion source and quadrupoles were 300, 300, and 150 °C, respectively. An electron-impact ion source at 70 eV was required. Nitrogen with a purity of 99.9% was produced by a NiGen LC-MS 40–1Claind generator (Gengaz, Wasquehal, France) and used as collision gas at 1.5 mL min^−1^. Appendix A shows the optimized parameters in Multiple Reaction Monitoring mode.

#### 2.3.2. Experiment Validation

A calibration graph was established for each of the 5 compounds using spiked uncontaminated soils at 7 increasing concentrations (2.5, 5, 10, 25, 50, 100 and 200 ng g^−1^). The spiked calibration samples were treated following the QuEChERS extraction procedure detailed above. Calibration curves were constructed by plotting the analyte/tebuconazole d6 peak area ratios versus the triazole concentration. Five experimental replicates for each concentration level were carried out. Validation of the method was studied through the following parameters: linearity (coefficient of determination), limit of quantification (QL) calculated at a signal-to-noise ratio of 10, precision expressed by the relative standard deviation (RSD) for concentration levels and trueness expressed as average recoveries of concentration levels.

The soils sampled in the two regions, Lorraine and Burgundy, were extracted and analysed under the conditions previously described. Some of the plots had been treated with metconazole and prothioconazole; the analysis of these molecules was performed by the Eurofins platform (Saverne, France). The limit of quantification for metconazole and prothioconazole were 50 ng/g, a concentration much higher than the limits of quantification achieved with the technique developed in our laboratory (see Results section).

### 2.4. Isolation and Identification of A. fumigatus from Soils

A 2 g subsample was removed from each soil sample and suspended in 10 mL of 0.1% Tween 80 solution (Merck^®^, Darmstadt, Germany). The suspension was vortexed for 1 min and 100 µL was used to seed each of three plates containing different culture media: dichloran-glycerol agar (DG18, Oxoid, Basingstoke, Hampshire, UK), and two malt agars supplemented with 2 mg/L itraconazole (called Maltitra) or 1 mg/L voriconazole (called Maltvori) [21,22]. The plates were incubated at 48°C for 48 h to promote the growth of *A. fumigatus* to the detriment of its cryptic species [23]. *A. fumigatus* isolates were identified on the basis of their macroscopic and microscopic characteristics. The DNA of isolates was extracted using the DNeasy Plant Mini Kit™ (Qiagen^®^, Milan, Italy), and a part of the *beta-tubulin* gene was amplified by PCR with primers Bt2a GGTAACCAAATCGGTGCTGCTTTC and Bt2b ACCCTCAGTGTAGTGACCCTTGGC [24] and sequenced (Sanger Sequencing, Microsynth platform, Balgach, Switzerland).

The susceptibility to itraconazole (ITZ) and voriconazole (VRC) of the isolates obtained was determined by the EUCAST (European Committee on Antimicrobial Susceptibility) microdilution method. EUCAST breakpoints were used for the interpretation of minimal inhibitory concentration (MIC), with an MIC > 1 mg L^−1^ for ITZ and VRC considered indicative of azole resistance [25]. In each EUCAST plate, two *A. fumigatus* were used as references: a sensitive strain (CBS 101355, MIC = 1 mg/L for ITZ and VRC) and an environmental strain with the TR34/L98H mutation, pan-resistant to antifungal triazoles (MIC > 16 mg/L for ITZ and MIC = 4 mg/L for VRC).

### 2.5. Assessment of Fungal Load by qPCR

Total DNA from the composite samples of each plot (pool of 10 sampling points for mycology analysis) was extracted with the Nucleospin soil kit (Macherey-Nagel^®^, Düren, Germany). Two TaqMan hydrolysis-quantitative real-time PCR (qPCR) assays were used to quantify *A. fumigatus* DNA [26] and the total levels of fungal species [27]. For the assay targeting *A. fumigatus*, the probe, forward and reverse primers were: FAM- CCCGCCGAAGACCCCAACATG-TAMRA, GCCCGCCGTTTCGAC and CCGTTGTTGAAAGTTTTAACTGATTAC. For the pan-fungal assay, the probe, forward and reverse primers were: GGRAAACTCACCAGGTCCAG, GSWCTATCCCCAKCACGA, FAM-TGGTGCATGGCCGTT-MGBNFQ.

The qPCR reactions were performed in a final volume of 20 μL (10 μL Gene Expression Master Mix (ThermoFisher Scientific, Waltham, MA, USA), 5 μL DNA, 1000 nM primers, 1000 nM probe in DNA-free water) with the Applied Biosystems 7500 Fast System (Life Technologies, Carlsbad, CA, USA), using the following program: 10 min at 95 °C, 45 cycles of 15 s at 95 °C and 1 min at 60 °C. A random 167 bp nucleotide sequence was used as an internal control for inhibition of the reaction [28]. Ten-fold dilutions of *A. fumigatus*-calibrated DNA (from 5000 fg/µL to 0.5 fg/µL) were amplified for each qPCR run to quantified targeted DNA samples. Nuclease-free water was used as a negative control in each series of amplification.

A retrospective analysis of soil samples from flowerbeds was also performed [29]. Soil sampling in the flowerbeds around the Besançon hospital was carried out in 2019. Seventy-one percent of the *A. fumigatus* isolated by culture was resistant to azoles and carried the TR_34_/L98H mutation in their *Cyp51A* gene. These isolates were mainly found in pots containing tulip bulbs from the Netherlands and treated with triazoles. DNA was extracted as previously described from 7 flowerbeds soils and the same qPCR (pan fungal and targeted *A. fumigatus*) was performed to compare field samples with areas containing ARAf. Among the 7 flowerbeds soil samples, resistant strains were found in 6/7 samples and the remaining sample (A22 in Table 2) had no resistant strains.

### 2.6. Statistical Analysis

The data were represented and analysed with Rstudio software (3.3.0 version, Boston, MA, USA). Pearson’s correlation assessments were performed to determine the linear relationship between the results of the two qPCR assays (panfungal and *A. fumigatus*-specific). Non-parametric tests (Wilcoxon or Kruskal-Wallis tests) were used for comparisons of total fungus concentrations (quantified by panfungal qPCR) or *A. fumigatus* concentrations (quantified by culture or qPCR) by region, type of culture and fungicide treatment. Spearman’s rank correlation assessments were performed to assess the relationship between the physical and chemical characteristics of the soil (OM content, pH, amounts of azoles and proportions of clay, silt and sand) and fungal load.

## 3. Results

### 3.1. Physicochemical and Fungicide Analyses of Soils

The soils collected in Lorraine and Burgundy differed little in terms of organic matter content, with mean values of 6.1% for each region and values ranging from 3.4% to 9.1% in Burgundy and 3.7 to 10.1% in Lorraine (Table 1). Slightly larger differences were observed for pH, with mean values of 6.5 for Lorraine and 7.5 for Burgundy (Table 1). Texture was assessed with the Aisne (15 classes) triangle (Figure 1). The vast majority of soils in the two regions (19/22) had a clay–silt composition. The three exceptions were two sandy loam soils in Lorraine and one sandy clay soil in Burgundy.

Calibration standard solutions of soil matrices at seven different levels provide a linear relationship between triazole/tebuconazole d6 peak area ratios and triazole concentrations with determination coefficients higher than 0.9972 for all analytes (Table 3). Limits of quantification were in the range of 2.5–13.8 ng/g; values were significantly lower than those obtained for metconazole. Precision gave good values for all compounds (less than or close to 20%) and trueness was situated in the range of 91–107% (acceptable values).

Tebuconazole was found in all field-crop soils analysed, at concentrations ranging from 5.6 ng/g to 19.1 ng/g of soil (Table 1). The soil with the highest level of tebuconazole contamination came from a field that had been treated with this molecule in the year of sampling. Residue levels were lower in the soils from other fields that had been treated with tebuconazole. This molecule was even found in organically farmed soils, which had not been treated with any azole fungicide for more than 14 years at the time of sampling. Epoxiconazole residues were also found in 5 of the 11 plots sampled in Lorraine. Metconazole, difenoconazole, propiconazole bromuconazole and prothioconazole were not found in any of the samples.

### 3.2. Fungal Analysis of Soils

In total, 166 fungal isolates were identified as belonging to the *A. fumigatus* complex, with 0 to 36 such isolates obtained per sample on three agar plates (DG18, MaltItra and MaltVori) (Table 1). The maximum quantity of isolates from *A. fumigatus* complex obtained in field crops was therefore 600 isolates per gram of soil. The mean and median amounts of *A. fumigatus* isolated were slightly higher for the Burgundy region, but this difference was not significant (Wilcoxon test, *p* = 0.26). In Lorraine, a mean of 100 colony-forming units (CFU) per gram of soil (with a median of 17 CFU/g of soil) was detected, whereas a mean of 15 CFU/g of soil was detected in Burgundy (with a median of 7 CFU/g of soil). Sixteen of the 166 isolates were detected on azole-containing media: 21 on MaltItra for the Lorraine region, 16 on MaltItra and 23 on MaltVori for the Burgundy region. Although 60 of the 166 isolates detected were isolated on azole media, none of the strains was found to be resistant. MIC are presented in Appendix A.

No qPCR inhibitors were detected in samples: positive control had a Cq of 34.2 and the average and median Cq for samples were 34.0 (SD = 0.2).

Cq obtained for standard curves, amplification efficiencies and the R^2^ for qPCR assays (pan-fungal target and specific *A. fumigatus* target) are presented Appendix A. Pan-fungal qPCR results were positive for all samples (minimal quantity = 0.4 ng/µL, median quantity = 4.1 ng/µL and maximal quantity = 10.4 ng/µL) and *A. fumigatus* qPCR was positive for 21/22 samples (median quantity = 8.3 fg/µL and maximal quantity = 61.4 fg/µL) (Figure 2). No correlation was found between the results of these two qPCR tests and no correlation was found between *A. fumigatus* load (fg/µL) using qPCR and *A. fumigatus* counts isolated in culture.

The mean fungal load detected by qPCR was similar in the two regions. However, the plot with the highest fungal load was located in the Lorraine region (plot CBP). It yielded 26 times more total fungal DNA than the plot with the lowest total fungal DNA, located in Burgundy (plot CHP).

### 3.3. Characterisation of the Fungal Environment

No significant difference in fungal load (pan-fungal qPCR or *A. fumigatus* detection by culture or quantification by qPCR) was found between the different treatments applied to field crops (Kruskal–Wallis tests: *p* = 0.26 for pan-fungal qPCR; *p* = 0.58 for *A. fumigatus* detection by culture, *p* = 0.61 for *A. fumigatus qPCR*). However, even though no significant difference was detected, organically farmed soils appeared to have higher median concentrations of total fungal DNA than soils grown under other treatment conditions (Figure 3). No difference in fungal load determined by culture or qPCR was found for any of the other parameters tested (region, type of crop, organic matter content, pH, texture or amounts of azoles; see Appendix A).

The mean fungal loads quantified by qPCR in flowerbeds was 121.9 ng µL^−1^ for the pan-fungal target and 67,644.1 fg/µL for the specific *A. fumigatus* target (Table 2). Thus, the amount of DNA for *A. fumigatus* was, on average, 5600 times lower in field-crop soils than in flowerbeds. Total fungal DNA levels were also, on average, 30 times lower in field-crop soils than in flowerbed soils.

## 4. Discussion

Our results indicate that field-crop soils in the French regions (Lorraine and Burgundy) studied are not currently hotspots of azole resistance in *Aspergillus fumigatus*. No resistant strains were found at any time (one month after azole fungicide treatment or four to five months after treatment), and the detected quantities of azoles fungicides were similar. In fact, azole fungicide residues were found in all the soils analysed, even those from plots on which these fungicides had not been used for years. The concentrations of these fungicides remained low, but they might nevertheless have long-term effects.

We investigated the physicochemical parameters of soils because they can directly influence the development of microbial populations and the diffusion of pesticides through the soil [30,31,32]. The texture analysis showed that the field-crop soils studied were similar, whereas we had expected to see differences, based on the information collected from farmers. However, one marked contrast was observed for pH and organic matter content, with soils of various degrees of acidity and organic matter content varying by a factor of two, with no impact on the presence of *A. fumigatus* and azole residues. Nevertheless, the texture, organic matter content and pH values obtained are similar to those previously reported for French agricultural soils [33]. We can therefore assume that the soils studied here correspond to the soil types classically found in field crops and that the lack of ARAf in this study does not result from plot selection bias. The percentages of organic matter observed here are higher than those reported in another study on vegetable crops in eastern France, in which ARAf isolates were obtained (maximum organic matter content of 4.5%) [34]. We would, therefore, have expected to find more *A. fumigatus* in field-crop soils if organic matter were the only factor driving the development of this species.

Data on crop treatment practices, particularly for the amount of fungicide applied and the date of treatments, are particularly difficult to obtain, and very few studies have combined attempts to detect ARAf with determinations of the concentrations of azole molecules present in the soil [34,35,36,37,38]. In our study, we only had access to two years of treatment history for Lorraine region, which does not allow us to fully explain our results, especially for epoxiconazole, which was detected in Lorraine soils but not reported in the treatment history provided. We found that all the soils tested contained tebuconazole, at a median concentration of 7.1 ng/g, even though this fungicide had not been applied for 14 years. Other azole molecules were not quantified, but this does not mean that they are not present, particularly metconazole and prothioconazole (quantification subcontracted to Eurofins), for which the limit of quantification is 50 ng/g.

Tebuconazole and epoxiconazole are the triazole molecules most frequently found in field-crop soils [39]. A large study of 317 agricultural soils in Europe quantified these molecules and reported a median concentration of 20 ng/g for tebuconazole and epoxiconazole. One study on 180 field-crop soils in France in 2021 reported that epoxiconazole was widely present, at a median concentration of 34.6 ng/g [40]. The maximum concentrations obtained for field-crop soils in these two studies were 130 ng/g for tebuconazole and 283 ng/g for epoxiconazole.

The concentrations obtained in our study were, thus, much lower than in other studies and were below the predicted environmental concentrations (PECs) in soils, such as 190–90 ng/g for tebuconazole at 0 and 100 days after treatment, respectively, for European cereals crops [39]. However, even if the concentrations detected were low and well below the PECs, the presence of these molecules in soils that have been farmed organically for more than 14 years reveals that diffusion may occur from neighbouring plots and/or that these molecules can persist for very long periods in the soil. Pesticide residues are globally a threat to the environment and human health, but they could also lead to ARAf in the long term, with the contamination of different environmental sections (air, soil, water and vegetation) [41]. Studies on experimental plots (where all treatment data are controlled) could be a plus to study the fate of azole fungicides in soils and the impact on resistance development.

Data on the presence of ARAf in field-crop soils are scarce. Most studies of this type are performed in connection with cases of aspergillosis in patients who may have been colonised through environmental contact with the fungus. In France, two studies have described resistant strains in cereal-crop soils: one of these studies investigated the environment close to the home of a patient and reported the presence of five triazole pan-resistant strains carrying the TR_34_/L_98_H mutation in barley fields [42]. However, nothing was known about the fungicide treatments used on these crops. Another study conducted in the environment of a patient who was also a farmer revealed the presence of one resistant strain carrying the TR_34_/L_98_H mutation among 128 strains analysed (0.8%). In this case, the field concerned had been treated with triazole molecules in the past but had been left fallow in the year of sampling [43].

We obtained 166 isolates from *A. fumigatus* complex, but none were resistant to the triazole molecules tested. A few studies in cereal crops in Europe have revealed relatively low resistance rates. In Germany, samples were taken from wheat and barley crops over three consecutive years: the percentages of ARAf present in these soils ranged from 0 to 3% between 2016 and 2018, with the TR_34_/L_98_H mutation being the main cause of resistance [18]. A study on soils from wheat fields in France, Germany and the UK was also conducted. It showed that 2 of the 418 strains tested for (0.5%) displayed pan-resistance to triazoles due to the TR_34_/L_98_H mutation, considering arable crop production a coldspot for azole resistance development [14]. These European resistance rates are lower than those reported for other countries. In comparison, ARAf rates of 8.6% and 12.3% were reported for rice fields in China and India, respectively [44,45]. The reported rates of ARAf are higher for other areas of professional activity, at 21% for vegetable cultivation and 14% for composting, for example [46,47], with some of these activities considered potential hotspots of resistance. A lack of dead-plant biomass, the preferred substrate for *A. fumigatus,* in field crops might account for this difference [13]. Even though no resistance was observed in this study and other authors have showed that low resistance is related to field agriculture, to neglect the monitoring of these environments might be dangerous as there is a possibility of identifying early resistance emergence and thus taking preventive actions. Moreover, it remains important to continue to be interested in the environment of patients at risk and to be vigilant in the management of patients if they are close to or work in professional environments where azole fungicides are used: this is the case of the two French studies mentioned above [42,43] and of other work carried out in our team such as sawmills [48]. In the context of the massive use of triazoles, the intrinsic resistance demonstrated, such as in some cryptic species of *A. fumigatus*, could represent a new threat if colonization by these species becomes more frequent. Research to define their distribution in the environment and their ecological niche and to determine their involvement in aspergillosis is needed. This is also the case for species of the section *Terrei* which in some regions show significant rates of resistance to posaconazole [49]. Finally, mucorales can also have a reduced sensitivity to triazoles (with a marked intrinsic resistance for some species), and it is therefore not impossible that these moulds can thrive in environments largely contaminated with triazoles. However, few environmental studies have focused on these species and their resistance to azoles.

In this study, we focused not only on checking for azole resistance in *A. fumigatus* in field-crop soils, but also the physicochemical characterisation of these soils. Differences in total fungal DNA levels between treated soils and soils managed under organic farming practices for a number of years suggested that repeated treatment is leading to a gradual depletion of fungal communities, particularly in Burgundy. Soil exposure to azoles has already been reported to lead to a decrease in microbial biomass affecting fungal populations in particular, with soil effects persisting for several months after treatment [50].

To our knowledge, no study has ever evaluated soil fungal communities, focusing particularly on *A. fumigatus*, by qPCR. We compared the results of this study with those of a previous study of flowerbed soils in which a large number of *A. fumigatus* isolates and a resistance rate of 71% were discovered [29]. The amounts of DNA for this species detected in field-crop soil were much lower than those in flowerbeds, by a factor of 5000.

In the field-crop soils, the maximum load was 600 CFU per gram of soil. In comparison, in the study by Schoustra et al., 2.9 × 10^4^ to 9.4 × 10^5^ CFU/g of soil (total *A. fumigatus*) were isolated from flower bulb waste, which was defined as a hotspot of resistance (with 180 to 2.3 × 10^5^ CFU of ARAf/g of soil) [13].

These results seem to confirm that the conditions in these field plots are not optimal for the development of this fungal species and resistant strains. However, it remains unclear how resistant fungal populations are maintained in flowerbeds containing azole-treated bulbs (not directly treated in the soil as field crops are), particularly as the resistant population disappears when the bulbs are replaced with organic bulbs [51].

Triazole-resistant strains of *A. fumigatus* have already been detected in field crops in Europe, but this phenomenon remains rarer in field crops than in other situations in which there may be residues of triazole molecules, such as compost, treated wood or waste from flower bulbs [13,52]. Field-crop soils do not appear to have the optimal characteristics for the development of *A. fumigatus* and azole resistance. *A. fumigatus* has been described as ubiquitous, but it clearly has environmental preferences. The ecology of this species and other fungal pathogens has been little studied, and further studies might improve the prediction of environments likely to become resistance hotspots.

## Figures and Tables

**Figure 1 jof-09-00618-f001:**
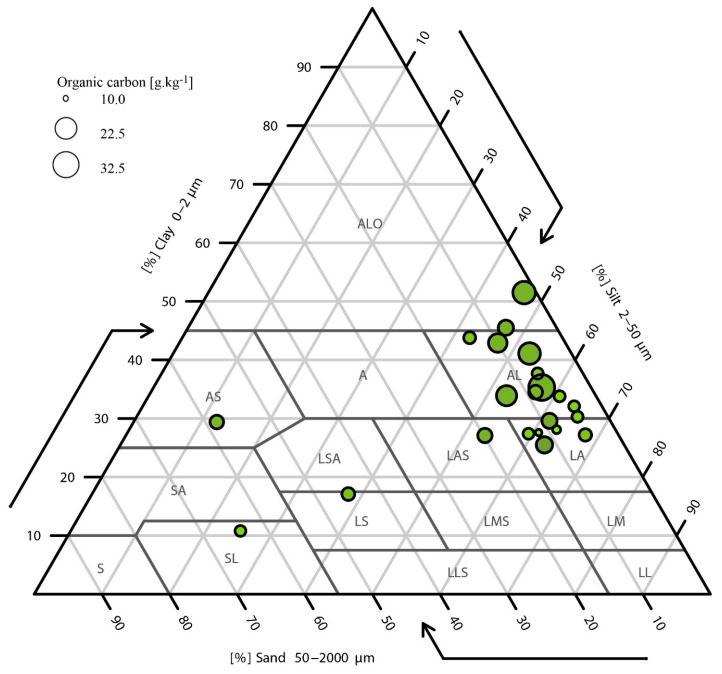
The French Aisne soil texture triangle and bubble diagram of total organic carbon for the 22 soils analysed. ALO: Heavy clay; SA: Clayey sand; LS: Sandy loam; AS: Sandy clay; SL: Silty sand; LMS: Medium sandy loam; A: Clay; LSA: Sandy clay loam; LM: Medium loam; AL: Silty clay; LAS: Sandy-clay loam; LLS: Light sandy loam; S: Sand; LA: Clay loam; LL: Light loam.

**Figure 2 jof-09-00618-f002:**
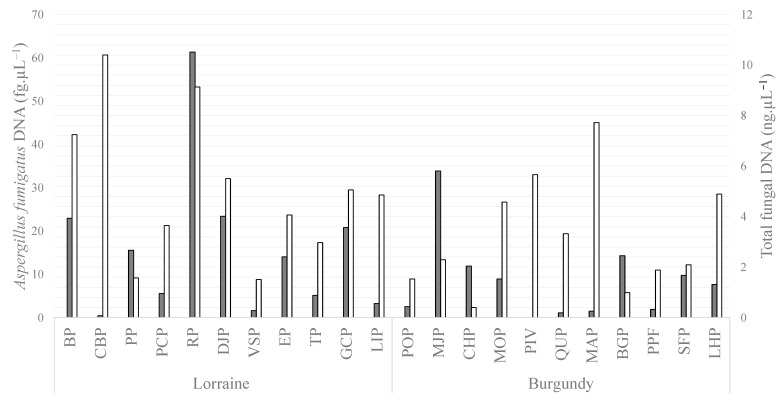
Amounts of DNA (mean of duplicates) found in field-crop soils from Lorraine and Burgundy by real-time quantitative PCR (qPCR) targeting *Aspergillus fumigatus* (in fg/µL) and targeting total fungal DNA (in ng/µL).

**Figure 3 jof-09-00618-f003:**
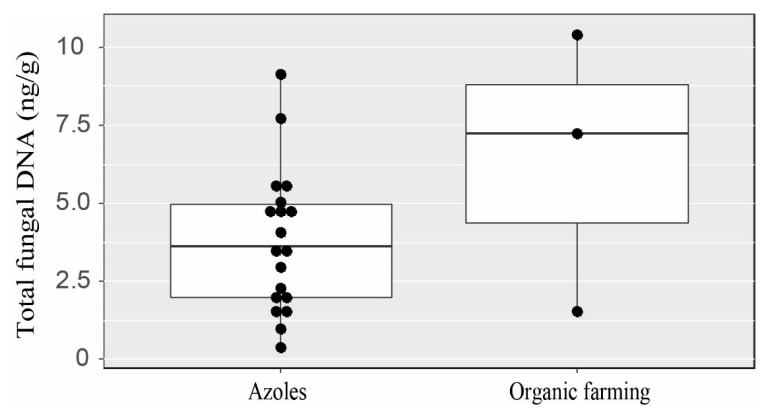
Amounts of total fungal DNA (in ng/g) detected by qPCR between field-crop soils treated with azole during the two previous years (*n* = 19) and organic farming soils (*n* = 3).

**Table 1 jof-09-00618-t001:** Characteristics of field crop soils and data obtained for the textural, chemical and mycological analysis of soils.

					**Structural Analysis**					**Chemical Analysis**		**Mycological Analysis**
	**Plot ID**	**Crop**	**Area (ha)**	**Azole Molecules Sprayed over the Two Years before Sampling**	**OM (%)**	**pH**	**Silt (%)**	**Sand (%)**	**Clay (%)**	**TBC (ng/g)**	**EPC (ng/g)**	** *Aspergillus fumigatus* ** **/g of Soil**
Lorraine	BP	wheat	3.0	Organic farming	6.2	7.9	56.9	8.6	34.5	6.0	<QL	17
	CBP	wheat	3.1	Organic farming	5.6	5.9	53.1	19.8	27.1	6.2	<QL	0
	PP	wheat	20.5	TBC, PTC, MTC	7.3	7.2	47.1	10	42.9	7.1	4.2	0
	PCP	wheat	9.3	TBC, PTC, MTC	4.5	5.9	37.9	45	17.1	7.1	<QL	17
	RP	wheat	12.6	TBC, PTC, MTC	6.3	6.3	61.4	9	29.6	8.1	4.6	17
	DJP	wheat	22.6	TBC, PTC, MTC	5.5	5.9	65.2	4.5	30.3	19.1	<QL	467
	VSP	wheat	7.9	TBC, PTC, MTC	3.7	5.2	63.8	4.1	32.1	7.0	3.0	367
	EP	wheat	7.2	TBC, PTC, MTC	5.1	6.2	67.9	4.9	27.2	6.6	3.4	0
	TP	wheat	1.5	TBC, PTC, MTC	10.1	6.8	46.7	1.8	51.5	6.2	<QL	150
	GCP	wheat	12.1	TBC, PTC	7.6	7.9	47	7.5	45.5	6.0	3.2	17
	LIP	wheat	7.3	TBC, PTC, MTC	4.9	6.6	25.1	64.1	10.8	7.0	<QL	33
Burgundy	POP	fallow	16.3	Organic farming	4.8	7.6	55.6	6.7	37.7	5.6	<QL	83
	MJP	maize	4.4	No azole	5.1	8	59.4	13.2	27.4	5.7	<QL	67
	CHP	maize	3.0	No azole	3.4	6.8	60.8	11.6	27.6	5.9	<QL	17
	MOP	maize	8.9	No azole	6.9	7.8	62.7	11.8	25.5	5.5	<QL	433
	PIV	wheat	16.0	PTC	4.3	6.9	63.2	8.7	28.1	5.9	<QL	0
	QUP	wheat	12.0	TBC, PTC, DFC	5.8	7.9	42.5	13.7	43.8	11.0	<QL	100
	MAP	wheat	6.4	TBC, PTC, DFC	5.1	6.5	60.8	5.4	33.8	10.7	<QL	50
	BGP	wheat	2.9	TBC, PTC	9.1	7.9	57.4	7.3	35.3	9.8	<QL	50
	PPF	oilseed rape	7.4	MTC, PTC (90 g/L)	7.4	7.6	52.7	6.2	41.1	8.1	<QL	0
	SFP	oilseed rape	16.3	MTC, PTC	7.7	7.7	52.9	13.2	33.9	12.2	<QL	600
	LHP	oilseed rape	11.5	MTC, PTC	7.6	7.7	12.3	58.3	29.4	14.7	<QL	83

OM: Organic matter; EPC: Epoxiconazole; TBC: Tebuconazole; PTC: Prothioconazole; MTC: Metconazole; DFC: Difenoconazole. QL = quantification limit (2.7 ng/g for tebuconazole and 2.5 ng/g for epoxiconazole).

**Table 2 jof-09-00618-t002:** qPCR results of flowerbeds soils and number of ARAf detected.

Samples	Number of ARAf per Sample (Culture Media)	*Aspergillus fumigatus* qPCR, Mean Quantity (ng/µL)	Pan Fungal qPCR, Mean Quantity (ng/µL)
A22	0	0.000959	270.5
D3	1	0.0246	74.85
D9	3	0.05935	44.1
D2	4	0.07415	49.3
D10	8	0.0943	98.1
A27	5	0.10265	113
D8	8	0.1175	203.5
median quantity (qPCR)		0.07415	98.1
mean quantity (qPCR)		0.0676441	121.9
minimal quantity (qPCR)		0.000959	44.1
maximal quantity (qPCR)		0.1175	270.5

**Table 3 jof-09-00618-t003:** Coefficient determination (r²), limit of quantification, (QL) relative standard deviation (RSD) and recovery for the determination of triazole.

	Determination Coefficient	QL (ng/g)	RSD (%)	Recovery (%)
Propiconazole	0.9987	9.0	17	102
Tebuconazole	0.9981	2.7	16	107
Epoxiconazole	0.9985	2.5	20	91
Bromuconazole	0.9983	7.6	21	94
Difenoconazole	0.9972	13.8	21	97

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
