# Peer review of "Field-Crop Soils in Eastern France: Coldspots of Azole-Resistant Aspergillus fumigatus"

_jof, 2023, doi:10.3390/jof9060618_

Round 1

Reviewer 1 Report

The manuscript: “Field crops: coldspots of azole-resistant Aspergillus fumigatus” describes the analysis of field samples for azole presence and the presence of Aspergillus fumigatus, and ARAf. This research is important to pinpoint the exact ecological niche of A. fumigatus and where azole resistance is likely to develop. While important, this manuscript raises questions about the data, the source of the samples and the analysis. All experiments need a more detailed description, and several key pieces of data are missing.

Major comments:

-       Quantification via mass spectrometry requires analytical standards to be analysed first for LOD, followed by a standard curve to quantify the samples. In addition, an internal control needs to be added to account for differences in extraction efficiency. None of this has been shown, or described. This is essential to be included.

-       The MIC data from the isolates is completely lacking and needs to be included. How is it possible that isolates were selected on azole-containing medium but none are considered resistant? That doesn’t make any sense.

-       Data on sampling sites, dates of sampling, dates of azole treatment and how long ago it was treated – all of this needs to be included.

-        

Other comments:

L2: Title does not reflect the data: Field crops were not sampled, but soils. Only fields in Eastern France were sampled and not a full year – seasonality can play a role.

L27-28: natural ecological niche is the soil. Actually data has proven (mycobiome and metagenomics) that A. fumigatus can barely be found in soil.

L36-38 Avoidance of triazoles as first-line treatment. Reference, I highly doubt this is true. First-line is still triazoles according to all guidelines.

L38-40 Resistance is more commonly associated with environmental use of azoles. Good example of population genomics from Rhodes et al 2022 and Snelders et al prove these points.

L40-43 More have been such as mefentrifluconazole, prothioconazole. Why was prothioconazole not measured? The fields have been treated with this? From MS data you can go back and reanalyse this, once a analytical standard has been run.

L44-45 Ref the WHO Fungal Priority List here. Araf was not classed as “at risk” but as critical.

L48-49 This definition is not really correct, because by this standard your soils are hotspots. Azole molecules detected + Afum can thrive. Completely negates this whole paper.

L49-51 Several more great papers show this.

L65 and Table 1: Essential to know when azoles were sprayed during the season – if long before your sampling then you would expect lower azole concentrations but Araf can develop due to low levels allowing selective evolutionary sweeps.

Table1: Are you sure only TBC was sprayed, so why is EPC found in some fields? Even TBC found in organic fields – how is this explained?

L95-99 To detect and quantify compounds, standards should be ran first as a standard curve. Were these performed and what did the data show? From this your LOD comes? In addition, was an internal standard included? Essential to quantify extraction efficiency to quantify the compounds in the sample?

L109-110: This is not standard practice for isolation of Afum? Adding azoles only allows Afum that is resistant to grow? This contradicts all the other data? Or did all isolates come from DG18 medium? Also the incubation at 48 Celsius selects for Afum that can withstand this temperature and not all isolates can? These limits need to be stated.

L112: How was DNA isolation and sequencing done? What primers? Where are the sequences? How analysed?

L113-117: EUCAST breakpoints need to include standards when running environmental isolates? Were these included? The MICs of the isolates need to be included too?

L119-122 How was ensured the amount of total DNA and Afum DNA quantified was not affected by extraction efficiency? Was there normalised for anything?

L129: Soil samples from flowerbeds, where is this data?

L164-165: Show this data then? What are the MICs? It needs to be made clear that they might be resistant to the environmental azoles but only voriconazole and itraconazole were tested for MICs? There needs to be an explanation why they grow on azole containing medium but are not resistant – this seems counterintuitive?

L166: No qPCR inhibitors? How was this shown? The data doesn’t explain this.

L166: Was the quantification of the qPCR performed using a standard curve of DNA? This should be explained better. Standard curves needs to be included.

L172: contaminated is not a good word to use for these fields. They were treated with azoles, or azole detected areas.

L177: give the exact P-value.

L177-179: Statistics on this? Else the statement should be removed.

L179: Contamination. Should be fungal load.

L179-181: How tested and what P-values. Even give if not significantly different.

L182: Where were these flowerbeds and what samples obtained? What was their treatment? Much more detail needed.

L182: Containing ARAf. How was this assessed? Where is this data? Everything needs to be included.

L189: Azole resistance development. This is not tested – only the presence of ARAf is tested. It could be that a year later there is ARAf.

L213: “contaminated” not contaminated, but contain.

L234-235 Most studies, but no refs supplied. Find population genomic studies that show environmental to patient transmission.

L259-260 This is not tested, active growth within the soil of A. fumigatus is not performed and could just be dormant spores present from air settling or even contamination of samples via the air.

Table 1: There seems to be a correlation between TBC/EPC and number of Afum found – has this been assessed?

Figure 2: How many replicates performed per sample? How was DNA calculated?

Figure 3: how many samples per treatment, would rather see the individual data points, these boxplots don’t tell us much. More fungal DNA in azole treated than untreated plots, how would this be described? What is the difference between untreated and organic farming plots – unclear?

Reviewer 2 Report

Dear Editor,

This study describe the investigation of 22 fields in two regions of Eastern France by looking at the A. fumigatus load and resistance fraction and also the soil structure and relevant fungicides in these fields.  Using qpcr to detect A. fumigatus load in soil is  a smart way. Author concluded crop field soil is cold spot based on the low amount of A. fumigatus was found and no resistant A. fumigatus was detected. Also confirmed by the comparison the qpcr results of hotspot (flowerbed) and crop field samples in this study.

But the import info is missing, what is the definition of cold spot, low number of A. fumigatus(how low)? low fraction of resistant A. fumigatus (how low?). Is there any breakpoint?? I think this need author illustrate more in intro

Hotspot are based on the present of azole and fungi can grow. In this study, author detected azoles and A. fumigatus, can we say this crop fields are hotspot as well? Can author illustrate more?

     Below are some comments

1.       L122,124. Please give the details over the methods how you qualify the total amount of fungi and  specific A. fumigatus (such as primers probe and primers used) rather than only listing references, it is convenient for reader.

2.       L159: author used CFU as the unit to present the A. fumigatus is impropriate. The general usge of CFU is combined with gram or ml. In this study, CFU/ gram would be better to describe the present of A. fumigatus, because the gram you used for analyse and how much liquid you used to dissolve are important factors for representing the real number of A. fumigatus. Here only absolute isolate account is available, how much gram is used, did not take into account. If there is max 36 colonies on the plate in 100ul, in total it is 1.8x103 CFU/ gram A. fumigatus in this sample.  Only mention 36 colonies rather than giving more info such as 2g misleads reader.

3.       L164-165: How can you explain “These 166 isolates were obtained on azole fungicide-containing me- 164 dium, but none of the strains was found to be resistant” does this mean the medium with fungicides doesnot select the resistance type or ???? “none of the strains was found to be resistant” is  confirmed by EUCAST?  Please add detailed info

4.       How many hospot samples (Flowerbed) were used for comparison? Did you have Fungal analysis of soils for flowerbed samples? (CFU data). It would be more convincing to include these data in the results section, rather than only one sentence in discussion.

5.       Is there any correlation between A. fumigatus load(fg/ul)  using qpcr and isolates counts of samples (CFU data)

Reviewer 3 Report

The manuscript (jof-2346211) entitled "Field crops: coldspot of azole-resistant Aspergillus fumigatus " investigated to determine whether large amounts of azole residues 56 and ARAf can be detected in field crops treated with azoles and to determine whether this environment constitutes a potential hotspot of resistance.

The manuscript is well written; the table is perfect. However, the manuscript requires revisions with gross English editing and confirmation for my questions.

Major Comments:

•           It is not clear what the study's clinical significance is - how does it help in clinical practice? The primary concern is that the impact of work is not presented enough in the manuscript – authors should at least speculate on the importance of their findings in practical application.

•           Common resistance mechanisms include mutations in the lanosterol 14 α-demethylase gene (CYP51A) that encodes the Cyp51A enzyme, the azoles' target, and tandem repeats (TR) in the promoter region of this gene. Although these resistance mechanisms have been described from broad geographical areas among both clinical and environmental isolates, the frequency of these resistance alleles varies considerably from country to country (<5%–30%). However, there may be other mechanisms of resistance associated with azole resistance, as some have reported that ~40% of azole resistance in A. fumigatus may be related to non-CYP51A mutations. What is your opinion about non-cyp51A azole-resistant strains??

The following article has been missed

Zoran T, Sartori B, Sappl L, Aigner M, Sánchez-Reus F, Rezusta A, Chowdhary A, Taj-Aldeen SJ, Arendrup MC, Oliveri S, Kontoyiannis DP, Alastruey-Izquierdo A, Lagrou K, Lo Cascio G, Meis JF, Buzina W, Farina C, Drogari-Apiranthitou M, Grancini A, Tortorano AM, Willinger B, Hamprecht A, Johnson E, Klingspor L, Arsic-Arsenijevic V, Cornely OA, Meletiadis J, Prammer W, Tullio V, Vehreschild JJ, Trovato L, Lewis RE, Segal E, Rath PM, Hamal P, Rodriguez-Iglesias M, Roilides E, Arikan-Akdagli S, Chakrabarti A, Colombo AL, Fernández MS, Martin-Gomez MT, Badali H, Petrikkos G, Klimko N, Heimann SM, Uzun O, Roudbary M, de la Fuente S, Houbraken J, Risslegger B, Sabino R, Lass-Flörl C, Lackner M. Corrigendum: Azole-Resistance in Aspergillus terreus and Related Species: An Emerging Problem or a Rare Phenomenon? Front Microbiol. 2019 Jan 14;9:3245. doi: 10.3389/fmicb.2018.03245. Erratum for: Front Microbiol. 2018 Mar 28;9:516. PMID: 30692970; PMCID: PMC6340063.

2.4: why did the authors not use the VIP medium comprising Sabouraud's dextrose agar plate (SDA; Difco), supplemented with 4 and 1 mg/L itraconazole and Voriconazole, respectively?

2 mg/L itraconazole (Maltitra) or 1 mg/L voriconazole (Maltvori)???

Colonies of A. fumigatus were recovered from the plates after two days of incubation.

Do you know exactly if you have recovered A. fumigatus or isolates from the Fumigati section? It is better to write A. fumigatus complex.

Susceptibility testing has been poorly written. Please add more informative data to the text.

•           In discussion,

pesticide behavior and development of azole-resistant isolates scenario have been missed, Vaezi A et al. Pesticide behavior in paddy fields and growth of azole-resistant Aspergillus fumigatus: Should we be concerned? J Mycol Med. 2018;28(1):59-64.

•           Nabili M, Shokohi T, Moazeni M, Khodavaisy S, Aliyali M, Badiee P, Zarrinfar H, Hagen F, Badali H. High prevalence of clinical and environmental triazole-resistant Aspergillus fumigatus in Iran: is it a challenging issue? J Med Microbiol. 2016 Jun;65(6):468-475. doi: 10.1099/jmm.0.000255. Epub 2016 Mar 23. PMID: 27008655.

Though the data seems interesting, the manuscript in its current form is tough to read and, therefore, to understand for the journal's readers (many samples collected at different places over time that have been exposed to various chemicals, etc..). A map showing the sampled sites would be helpful. The method used for in vitro susceptibility testing in the present study can also be considered a significant limitation.

This choice is unfortunate as some highly reproducible standardized techniques can be used, providing reliable data and allowing MIC comparisons with other studies.

-

Dear Editor

The manuscript (jof-2346211) entitled "Field crops: coldspot of azole-resistant Aspergillus fumigatus " investigated to determine whether large amounts of azole residues 56 and ARAf can be detected in field crops treated with azoles and to determine whether this environment constitutes a potential hotspot of resistance.

The manuscript is well written; the table is perfect. However, the manuscript requires revisions with gross English editing and confirmation for my questions.

Major Comments:

•           It is not clear what the study's clinical significance is - how does it help in clinical practice? The primary concern is that the impact of work is not presented enough in the manuscript – authors should at least speculate on the importance of their findings in practical application.

•           Common resistance mechanisms include mutations in the lanosterol 14 α-demethylase gene (CYP51A) that encodes the Cyp51A enzyme, the azoles' target, and tandem repeats (TR) in the promoter region of this gene. Although these resistance mechanisms have been described from broad geographical areas among both clinical and environmental isolates, the frequency of these resistance alleles varies considerably from country to country (<5%–30%). However, there may be other mechanisms of resistance associated with azole resistance, as some have reported that ~40% of azole resistance in A. fumigatus may be related to non-CYP51A mutations. What is your opinion about non-cyp51A azole-resistant strains??

The following article has been missed

Zoran T, Sartori B, Sappl L, Aigner M, Sánchez-Reus F, Rezusta A, Chowdhary A, Taj-Aldeen SJ, Arendrup MC, Oliveri S, Kontoyiannis DP, Alastruey-Izquierdo A, Lagrou K, Lo Cascio G, Meis JF, Buzina W, Farina C, Drogari-Apiranthitou M, Grancini A, Tortorano AM, Willinger B, Hamprecht A, Johnson E, Klingspor L, Arsic-Arsenijevic V, Cornely OA, Meletiadis J, Prammer W, Tullio V, Vehreschild JJ, Trovato L, Lewis RE, Segal E, Rath PM, Hamal P, Rodriguez-Iglesias M, Roilides E, Arikan-Akdagli S, Chakrabarti A, Colombo AL, Fernández MS, Martin-Gomez MT, Badali H, Petrikkos G, Klimko N, Heimann SM, Uzun O, Roudbary M, de la Fuente S, Houbraken J, Risslegger B, Sabino R, Lass-Flörl C, Lackner M. Corrigendum: Azole-Resistance in Aspergillus terreus and Related Species: An Emerging Problem or a Rare Phenomenon? Front Microbiol. 2019 Jan 14;9:3245. doi: 10.3389/fmicb.2018.03245. Erratum for: Front Microbiol. 2018 Mar 28;9:516. PMID: 30692970; PMCID: PMC6340063.

2.4: why did the authors not use the VIP medium comprising Sabouraud's dextrose agar plate (SDA; Difco), supplemented with 4 and 1 mg/L itraconazole and Voriconazole, respectively?

2 mg/L itraconazole (Maltitra) or 1 mg/L voriconazole (Maltvori)???

Colonies of A. fumigatus were recovered from the plates after two days of incubation.

Do you know exactly if you have recovered A. fumigatus or isolates from the Fumigati section? It is better to write A. fumigatus complex.

Susceptibility testing has been poorly written. Please add more informative data to the text.

•           In discussion,

pesticide behavior and development of azole-resistant isolates scenario have been missed, Vaezi A et al. Pesticide behavior in paddy fields and growth of azole-resistant Aspergillus fumigatus: Should we be concerned? J Mycol Med. 2018;28(1):59-64.

•           Nabili M, Shokohi T, Moazeni M, Khodavaisy S, Aliyali M, Badiee P, Zarrinfar H, Hagen F, Badali H. High prevalence of clinical and environmental triazole-resistant Aspergillus fumigatus in Iran: is it a challenging issue? J Med Microbiol. 2016 Jun;65(6):468-475. doi: 10.1099/jmm.0.000255. Epub 2016 Mar 23. PMID: 27008655.

Though the data seems interesting, the manuscript in its current form is tough to read and, therefore, to understand for the journal's readers (many samples collected at different places over time that have been exposed to various chemicals, etc..). A map showing the sampled sites would be helpful. The method used for in vitro susceptibility testing in the present study can also be considered a significant limitation.

This choice is unfortunate as some highly reproducible standardized techniques can be used, providing reliable data and allowing MIC comparisons with other studies.

Round 2

Reviewer 1 Report

The revisions in the manuscript have significantly improved this work and its ability to inform other researchers about the methodologies used, and to scrutinise the data in detail. I would like to commend the authors on their rigorous detailed descriptions that have been added. I only have some minor comments remaining:

- The order of Tables is out of order, Table 3 is in the text before Table 2.

- It would be easier to compare the same values in Table 2 ie put the A fumigatus ones in ng/uL as well.

- How is it possible that the concentration of Aspergillus qPCR is higher than the pan fungal qPCR. This can't be right. Is there anything different with the PCR efficiency?

- It is unclear how many data points are per category in Figure 3. Add this for each category. Preferably add the actual datapoints to the graph.

Author Response

- The order of Tables is out of order, Table 3 is in the text before Table 2.

Authors’ response:

Table 2 is mentioned line 198 “Among the 7 flowerbeds soil samples, resistant strains were found in 6/7 samples and the remaining sample (A22 in Table 2) had no resistant strains” and Table 3 line 222 “

So we have not made any changes.

- It would be easier to compare the same values in Table 2 ie put the A fumigatus ones in ng/uL as well.

Authors’ response:

We have expressed the results in the same unit in Table 2. On the other hand, for the figure it is not possible because the quantities of Aspergillus will not be visible on the graph if they are expressed in ng/µL. This may be confusing (see next comment of the reviewer) but we can't do otherwise.  

- How is it possible that the concentration of Aspergillus qPCR is higher than the pan fungal qPCR. This can't be right. Is there anything different with the PCR efficiency?

Authors’ response:

The concentrations of Aspergillus are not higher. The confusion comes from the fact that the units were not the same. Now that everything is expressed in ng/µL it should be easier to understand.

- It is unclear how many data points are per category in Figure 3. Add this for each category. Preferably add the actual datapoints to the graph.

Authors’ response:

We modified the graph using ggplot2 library in R.